# The Consequences of Soluble Epoxide Hydrolase Deletion on Tumorigenesis and Metastasis in a Mouse Model of Breast Cancer

**DOI:** 10.3390/ijms22137120

**Published:** 2021-07-01

**Authors:** Rushendhiran Kesavan, Timo Frömel, Sven Zukunft, Bernhard Brüne, Andreas Weigert, Ilka Wittig, Rüdiger Popp, Ingrid Fleming

**Affiliations:** 1Institute for Vascular Signalling, Centre for Molecular Medicine, Goethe University, 60590 Frankfurt, Germany; rushendhiran.kesavan@utsouthwestern.edu (R.K.); froemel@vrc.uni-frankfurt.de (T.F.); Svenzukunft@gmail.com (S.Z.); popp@vrc.uni-frankfurt.de (R.P.); 2Institute of Biochemistry I, Faculty of Medicine, Goethe University, 60590 Frankfurt, Germany; b.bruene@biochem.uni-frankfurt.de (B.B.); weigert@biochem.uni-frankfurt.de (A.W.); 3Functional Proteomics, Institute of Cardiovascular Physiology, Faculty of Medicine, Goethe University, 60590 Frankfurt, Germany; wittig@med.uni-frankfurt.de; 4German Centre for Cardiovascular Research (DZHK) Partner Site RheinMain, 60590 Frankfurt, Germany

**Keywords:** polyunsaturated fatty acid, cancer metastases, angiogenesis

## Abstract

Epoxides and diols of polyunsaturated fatty acids (PUFAs) are bioactive and can influence processes such as tumor cell proliferation and angiogenesis. Studies with inhibitors of the soluble epoxide hydrolase (sEH) in animals overexpressing cytochrome P450 enzymes or following the systemic administration of specific epoxides revealed a markedly increased incidence of tumor metastases. To determine whether PUFA epoxides increased metastases in a model of spontaneous breast cancer, sEH^-/-^ mice were crossed onto the polyoma middle T oncogene (PyMT) background. We found that the deletion of the sEH accelerated the growth of primary tumors and increased both the tumor macrophage count and angiogenesis. There were small differences in the epoxide/diol content of tumors, particularly in epoxyoctadecamonoenic acid versus dihydroxyoctadecenoic acid, and marked changes in the expression of proteins linked with cell proliferation and metabolism. However, there was no consequence of sEH inhibition on the formation of metastases in the lymph node or lung. Taken together, our results confirm previous reports of increased tumor growth in animals lacking sEH but fail to substantiate reports of enhanced lymph node or pulmonary metastases.

## 1. Introduction

Angiogenesis and inflammation are hallmarks of cancer that are known to be influenced by arachidonic acid and other polyunsaturated fatty acids (PUFAs). PUFAs are primarily metabolized by three distinct enzymatic systems, i.e., by cyclooxygenases, lipoxygenases, and cytochrome P450 enzymes (CYPs), to generate bioactive lipid mediators. Most is known about the prostanoids, leukotrienes, hydroxyeicosatetraenoic acids, and epoxyeicosatrienoic acids (EETs) generated from arachidonic acid, but the CYP enzymes can metabolize several different PUFAs to epoxides or ω-hydroxides with biological activity. As several CYP-derived PUFA metabolites have been implicated in inflammatory processes, they are widely studied as therapeutic targets of inflammation as well as anti-cancer targets. However, although the eicosanoids were first linked to endothelial cell proliferation and angiogenesis almost two decades ago [1,2], it is only recently that PUFA metabolites have been recognized as important signaling mediators in physiological and pathological angiogenesis (for review, see reference [3]).

The PUFA epoxides, and particularly the EETs, which are epoxides of arachidonic acid, have been associated with effects in solid tumors as well as in malignant hematological diseases. For example, CYP2C9 is the most abundantly expressed CYP enzyme in several human malignant neoplasms [4] and esophageal adenocarcinoma [5]. CYP2J2 is also upregulated in numerous forms of cancer [6] and has been detected in five human-derived malignant hematological cell lines, as well as in leukemia cells from peripheral blood and bone marrow in 86% of patients with malignant hematologic diseases [7,8]. In line with this, selective inhibitors of CYP2J2 were reported to exhibit strong activity against human cancers in vitro and in vivo [9]. 

The levels of PUFA epoxides in cells and tissues are usually tightly controlled by the activity of the soluble epoxide hydrolase (sEH), which metabolizes them to vicinal diols [3]. Thus, a decrease in sEH expression, as has been observed in some tumors [10,11,12], would be expected to shift the epoxide–diol balance towards that of the epoxides. Such considerations led to the proposal that increasing epoxide levels could predispose patients to tumor development and that this process could be promoted and maybe even accelerated following the inhibition or downregulation of the sEH. However, some of the previous studies in mice may not truly represent the physiological situation as they were performed by injecting cancer cells into animals overexpressing the human CYP2C8 or CYP2J2 enzymes in endothelial cells and a subset (Tie2-expressing) of myeloid cells, or following the systemic administration of high concentrations of 14,15-EET [7,13]. The aim of this investigation was to study the consequences of interfering with sEH expression on tumor growth and metastasis in genetically modified animals that spontaneously generate tumors without the exogenous application of high concentrations of epoxide mediators or inhibitors. Therefore, breast cancer development was studied in mice expressing the polyoma middle T oncogene (PyMT) under the control of the mouse mammary tumor virus promoter, to induce spontaneous mammary tumors [14]. To facilitate the study of endogenous sEH in tumor growth and angiogenesis, PyMT mice were then crossed with sEH^-/-^ mice to generate sEH-deficient mice that spontaneously generate breast tumors (so called PyMT^Δ^^sEH^ mice).

## 2. Results

### 2.1. Tumor Development and Metastases in PyMT and PyMT^ΔsEH^ Mice

Palpable tumors developed in adult PyMT^ΔsEH^ mice from 13 weeks of age and increased in number and size over the observation period (20 weeks). Compared with the PyMT mice, the number of tumors that developed, as well as their size and the overall tumor burden per mouse, were significantly greater in animals lacking sEH (Figure 1).

The inhibition and deletion of the sEH has been linked with escape from tumor dormancy and increased metastasis [13]. Therefore, the numbers of metastases that developed in axillary lymph node and lungs from the affected 20-week-old animals were determined. In contrast to the previous report, lymph node metastases (cytokeratin+, CD326+ cells) were comparable in PyMT and PyMT^ΔsEH^ mice (Figure 2a). Moreover, in the lung, the total number and the size of metastases did not differ between the two strains (Figure 2b).

### 2.2. Cellular Composition of Primary Tumors from PyMT, PyMT^2c44^, and PyMT^sEH^ Mice

Next, the formation of capillaries and lymph capillaries was assessed in primary tumors as a number of sEH substrates and products have been reported to stimulate angiogenesis (for review, see [15]). There was a clear difference in the vascularization of the PyMT and PyMT^ΔsEH^ tumors as the latter contained more vascular endothelial cells (endomucin+) than tumors from PyMT mice, as assessed using immunohistochemistry (Figure 3a). A similar phenomenon was detected when FACS analysis was used to identify vascular endothelial cells (CD31+, CD146+, Ly-6C+ cells) in primary tumor digests (Figure 3b).

Although immunohistochemistry suggested that more lymphatic endothelial cells (Lyve1+) were present in tumors from PyMT^ΔsEH^ mice, no significant differences in CD31+, CD146-, or Ly-6C- cells could be detected in tumor lysates. A comparison of the expression of vascular endothelial cell and lymph endothelial cell marker genes also confirmed the high angiogenic capacity of the PyMT^ΔsEH^ tumors, with more pronounced effects on blood endothelial cells (expressing Vegfa, Vegfr1, Vegfr2, Ephb4, and Sox7) than on lymphatic marker genes (Figure 3c). Thus, the increase in primary tumor size observed in PyMT^ΔsEH^ mice was associated with increased angiogenesis.

Primary tumors also displayed differences in inflammatory cell infiltration as more F4/80-expressing cells (macrophages) were detected in tumors from PyMT^ΔsEH^ mice (Figure 4a).

There was a tendency for the numbers of CD11b+ cells (myeloid cells) to be increased in tumors from PyMT^ΔsEH^ mice that failed to reach statistical significance. Moreover, quantification by FACS revealed the presence of more F4/80+ and Ly-6G+ (neutrophils) cells in the PyMT^ΔsEH^ tumors (Figure 4b). The higher macrophage and neutrophil content of tumors was reflected in the higher number of circulating monocytes and neutrophils in 20-week-old PyMT^ΔsEH^ mice (Appendix A). Interestingly, the platelet count was also higher in blood from PyMT^ΔsEH^ versus PyMT mice.

### 2.3. PUFA-Derived Lipid Metabolite Profiles in Primary Tumors from PyMT and PyMT^sEH^ Mice

The sEH metabolizes PUFA epoxides to their corresponding vicinal diols. Therefore, the concentrations of ω-6 and ω-3 PUFA epoxides and diols were assessed in primary tumors from PyMT and PyMT^ΔsEH^ mice. The deletion of the sEH did result in subtle changes in the tumor concentration of the arachidonic acid-derived diols 8,9-, 11,12-, and 14,15- dihydroxyeicosatrienoic acid (DHET) without significantly increasing the levels of the corresponding epoxides (Figure 5). However, concentrations of the linoleic acid-derived epoxides 9,10- and 12,13- epoxyoctadecamonoenic acid (EpOME) were elevated and were present in concentrations a magnitude higher than the levels of the EETs (i.e., 1–8 µg/g tissue versus 15–37 ng/g tissue).

There were also changes in the epoxides and diols derived from ω-3 PUFAs, with levels of 8,9-, 17,18-epoxyeicosatetraenoic acid (EEQ) and 19,20-epoxydocosapentaenoic acid (EDP) being higher in tumors from PyMT^ΔsEH^ versus PyMT mice (Figure 6). 

The consequences of sEH deletion on the levels of diols was less clear as 11,12-DHET and 17,18- dihydroxyeicosatetraenoic acid (DiHETE) concentrations increased while that of 13,14-dihydroxydocosapentaenoic acid (DHDP) decreased. Crosstalk between the cyclooxygenase and sEH enzymes has been proposed, and the lack of the sEH was also associated with an increase in 6-keto prostaglandin (PG) F1α, PGD_2_, and PGF_2__α_ levels (Figure 7). However, there was no difference in PGH_2_ or 12(S)-hydroxyheptadecatrienoic acid (12-HHT) levels in tumors from PyMT and PyMT^Δ^^sEH^ mice, a situation that contrasts starkly with the situation in PyMT mice that lack Cyp2c44 [16].

### 2.4. Consequences of sEH Deletion on the PyMT Tumor Proteome 

To gain insight into the consequences of sEH deletion on the tumor proteome, protein expression was compared in similarly sized primary tumors from PyMT and PyMT^ΔsEH^ mice. This revealed 20 proteins significantly elevated and 103 proteins significantly attenuated in tumors from PyMT^ΔsEH^ versus PyMT mice (Figure 8a, Dataset 1). There was no compensatory upregulation of either the microsomal epoxide hydrolase or the leukotriene A4 hydrolase in tumors from PyMT^ΔsEH^ versus PyMT mice. Many of the proteins downregulated in the absence of the sEH were related to RNA structure and processing as well as extracellular matrix and cell junction assembly (Figure 8b), while proteins elevated in tumors from PyMT^ΔsEH^ mice were involved in amino acid metabolism and the tricarboxylic acid cycle (Figure 8c). It was possible to confirm the decrease in the expression of carnitine palmitoyltransferase 1A (CPT1A) in tumors from PyMT^ΔsEH^ mice (Figure 8d). 

## 3. Discussion

The results of the present investigation revealed that in a mouse model of breast cancer, the deletion of the sEH accelerated the growth of primary tumors. This effect was paralleled by an increase in tumor angiogenesis as well as with an increase in tumor macrophages and neutrophils. In contrast to previous studies that assessed tumor development and metastasis in models relying on the injection of tumor cells, there was no consequence of sEH deletion on the development of metastases in the lymph nodes and lung in PyMT mice.

PUFA epoxides, such as the EETs, have been reported to exhibit anti-inflammatory properties in different cell and animal models [3]. Once the sEH was identified as the enzyme that largely determined the half-life of PUFA epoxides, there was a great deal of interest in developing inhibitors of the sEH to increase epoxide levels and maximize their beneficial properties [17,18]. Although there is experimental evidence that links sEH inhibition with improvements in inflammatory conditions, blood pressure, metabolic diseases, and even neuropathic pain [19], the impetus for the further development of such compounds was dampened by reports of a link between sEH inhibition and accelerated tumor growth [6,7,9,20] as well as increased metastases and escape from dormancy [13,21]. Despite the reports linking decreased sEH expression or activity to accelerated tumor growth and dissemination, there is also evidence for a protective role of sEH inhibition in carcinogenesis. For example, sEH deficiency inhibits sodium dextran sulfate-induced colitis and carcinogenesis in mice [22]. Moreover, the multi-kinase inhibitor sorafenib, which is used to treat some forms of cancer, is an effective sEH inhibitor—a factor that contributes to its actions in vivo [23]. Because many of the animal studies performed to investigate the role of PUFA metabolites in tumor growth relied on the manipulation of CYP enzyme levels and/or the exogenous application of EETs, as well as the injection of tumor cell lines, this study set out to assess the consequences of sEH deletion in a spontaneous model of breast cancer, i.e., in PyMT mice. This approach revealed that the development of breast cancer tumors was accelerated in PyMT^ΔsEH^ versus PyMT mice, with the PyMT lagging roughly 3 weeks behind their sEH-deficient littermates. This translated into a significantly increased overall tumor weight and tumor burden in 20-week-old animals. The accelerated growth of primary tumors in PyMT^ΔsEH^ mice was accompanied by an increase in tumor vascularization that fit with the well-described effects of epoxides of arachidonic acid on angiogenesis [24]. Rather surprisingly, given the previous reports that linked increased CYP enzyme and decreased sEH expression with more aggressive breast cancer tumors in humans [12], as well as a significant increase in tumor metastasis in sEH-deficient mice [13,22], we detected no effect of sEH deletion on the spreading of the primary tumor to the lymph nodes or the lung. These findings contrasted with the consequences of deleting Cyp2c44 on the same PyMT background. Indeed, tumor growth in PyMT mice lacking Cyp2c44 was clearly accelerated and associated with a pronounced increase in lung and lymph node metastasis [16]. It is possible that the differential effect on metastasis can be accounted for by the fact that sEH deletion, in contrast to the deletion of Cyp2c44, did not markedly stimulate lymphangiogenesis [16], which seems to be required for metastasis in PyMT mice [25].

Although the differences in tumor growth were relatively small, clear differences in protein expression were detected in primary tumors from PyMT and PyMT^ΔsEH^ mice. Proteins affected by sEH deletion in the tumors included the CYP oxidoreductase, which is not only required for CYP activity but has also been linked with ferroptosis [26,27]. Even though angiogenesis was enhanced in tumors lacking the sEH, few of the most significantly altered proteins could be directly linked with angiogenesis. Exceptions were α6 integrin and the venous marker and regulator of arteriovenous specification, EPH receptor B4 [28], which was previously reported to be regulated by 11,12-EET [29]. A-kinase anchoring protein (AKAP) 9 was also affected by sEH deletion but, while some AKAP proteins have been linked with angiogenesis [30], AKAP9 has stronger links with breast cancer cell migration [31,32,33]. Many of the proteins expressed at significantly lower levels in tumors lacking the sEH were linked to metabolism and included pyrroline-5-carboxylate reductase 2, which is involved in proline synthesis and leads to cancer cell proliferation [34]; 6-pyruvoyl tetrahydrobiopterin synthase; and aspartate aminotransferase, the ratio of which, relative to alanine aminotransferase, generates the De Ritis ratio [35], that is associated with poorer prognosis in patients [36]. There were also changes in the expression of phosphomannomutase 2, lactate dehydrogenase A chain, succinyl-CoA ligase, very long-chain specific acyl-CoA dehydrogenase, aconitate hydratase, argininosuccinate synthase, pyruvate kinase, glutathione peroxidase 1, glutathione S-transferase theta 2B, and CPT1A. We focused on the latter protein as, in addition to its role in metabolism, it has also been implicated in lymphangiogenesis [37]. In line with the lack of pronounced lymphangiogenesis in the tumors studied, CPT1A levels were lower in tumors from PyMT^ΔsEH^ than in PyMT mice. 

What then is the role of the sEH in cancer? One reason that the situation is not clear-cut is that the PUFA levels in a given tissue, together with the expression of the CYP enzymes, determine the epoxide profile that can modify cell and organ function. This means that, while inhibition of the sEH could increase ω-6 PUFA-derived EETs to promote angiogenesis in the one tissue, in another, where ω-3 PUFA epoxides dominate, the lack of the sEH retards angiogenesis. Such situations have been described for the heart, where EETs stimulate angiogenesis [38], and in the retina, where sEH inhibition results in attenuated angiogenesis because of the loss of the ω-3-PUFA-derived lipid mediator 19,20-dihydroxydocosapentaenoic acid [39]. The relevance of such a consideration was highlighted by the fact that tumor growth and metastasis were reduced in mice that received a sEH inhibitor as well as a diet rich in ω-3 PUFA to shift the PUFA epoxide balance more towards ω-3 metabolites [10]. Moreover, epoxides derived from the ω-3 PUFA, docosahexaenoic acid, were reported to inhibit angiogenesis, tumor growth, and metastasis [40], in exactly the same models in which 14,15-EET had the opposite effect [13]. Additional possibilities include crosstalk between the CYP-sEH pathway and other PUFA-metabolizing enzymes. Indeed, the angiogenesis stimulated by applying 8,9-EET to human aortic endothelial cells was attributed to its metabolism to 11-hydroxy-8,9-EET by COX2 [41]. In the current study, the lack of sEH in the PyMT^ΔsEH^ mice was linked to an increase in 6-keto PGF_1α_, a hydrolysis product of prostacyclin, PGF_2__α_, and PGD_2_. Prostacyclin is potentially interesting as the balance of prostacyclin and thromboxane A_2_ can result in the development of vascular endothelial dysfunction, which facilitates the progression of metastasis [42]. However, the prostaglandins can elicit inflammatory and anti-inflammatory actions [43] and, while PGD_2_ was previously linked with protective, anti-tumor effects [44], PGF_2__α_ could support tumorigenesis by creating an inflammatory environment [45]. There was no evidence of an effect of sEH deletion on the primary tumor levels of PGE_2_ or 12-HHT, a shift that characterized increased tumorigenesis and metastasis in PyMT mice lacking Cyp2c44 [16]. Preventing crosstalk between the CYP-sEH and COX pathways could account for the beneficial effects of treating PyMT mice lacking Cyp2c44 with celecoxib [16], as well as the benefits of dual sEH and COX inhibition on tumor growth and metastasis [46]. 

Taken together, our results confirm previous reports of increased tumor growth in animals lacking sEH, an effect that may be attributed to the increased angiogenetic potential of sEH-deficient tumors. However, in contrast to models in which cancer cells were injected into recipient mice lacking the sEH or treated with sEH inhibitors, we found no evidence of accelerated or enhanced lymph node or pulmonary metastases. 

## 4. Materials and Methods

### 4.1. Materials

Firstly, 11,12-Epoxyeicosatrienoic acid (11,12-EET) 11,12-dihydroxyeicosatrienoic acid (11,12-DHET), 14,15-epoxyeicosatrienoic acid (14,15-EET), 14,15-dihydroxyeicosatrienoic acid (14,15-DHET), 9,10-epoxyoctadecamonoenic acid (9,10-EpOME), 9,10-dihydroxyoctadecenoic acid (9,10-DiHOME), 12,13-epoxyoctadecamonoenic acid (12,13-EpOME), 12,13-dihydroxyoctadecenoic acid (12,13-DiHOME), prostaglandin (PG) E_2_, PGD_2_, PGI_2_, PGH_2_, 12(S)-hydroxyheptadeca-5Z,8E,10E-trienoic acid (12-HHT), malondialdehyde (MDA), 19,20-epoxydocosapentaenoic acid (19,20-EDP), and 19,20-dihydroxydocosapentaenoic acid (19,20-DHDP) were from Cayman Chemical (Ann Arbor, MI, USA). Acetonitrile (ultra LC–MS grade), ethanol (HPLC gradient grade), formic acid (p.a grade), and isopropanol (LC–MS grade) were from Roth (Karlsruhe, Germany), and ethyl acetate (LC grade) was from Merck (Darmstadt, Germany). All other compounds including flufenamic acid were from Sigma–Aldrich (Steinheim, Germany). 

### 4.2. Animals

Floxed sEH mice (Ephx2^tm1^.^1Arte^) were generated by TaconicArtemis GmbH (Cologne, Germany) as described [39] and crossed with Gt(ROSA)26Sortm16(Cre)Arte mice expressing Cre under the control of the endogenous Gt(ROSA)26Sor promoter (TaconicArtemis) to generate mice globally lacking sEH (sEH^-/-^). Polyoma virus middle T-antigen (PyMT) mice (FVB/N-Tg(MMTV-PyVT)634Mul/J) were purchased from The Jackson Laboratory (Bar Harbor, ME, USA) and then crossed with sEH^-/-^ mice to generate breast cancer mouse models lacking sEH (PyMT^ΔsEH^). Mice were housed in conditions that conformed to the Guide for the Care and Use of Laboratory Animals published by the U.S. National Institutes of Health (NIH publication no. 85–23). All experiments were approved by the governmental authorities (Regierungspräsidium Darmstadt: FU_1072). Female animals were used throughout and were screened from week 6 after birth for breast tumors. Tumor size (by caliper) and localization were monitored. For the isolation of tumors and organs, mice were killed using 4% isoflurane in air and subsequent exsanguination. This study was performed simultaneously with a study to assess the consequences of Cyp2c44 on tumor formation and metastasis in PyMT mice [16]. In accordance with the 3R principles of animal research, the same PyMT group was used for both studies.

### 4.3. Animal Monitoring and Tumor Biometrics

Tumor growth was monitored to a maximal tumor diameter of 1.5 cm and tumor volume was calculated using the formula: volume = length × width^2^ × 0.52. Lymph node metastases were determined by cytokeratin+ CD326+ cells in 5 sequential slides per animal. 

Lung metastases were determined in samples from 20-week-old mice using Meyer’s haemalum (Merck, Darmstadt, Germany) staining. The appearance of metastases (micro and macro) was evaluated in right lung sections of 5 sequential slides per animal. 

### 4.4. Immunohistochemistry

Primary tumors, axillary lymph nodes, and lungs were fixed in zinc fixative buffer (overnight, 4 °C) and 10-μm-thick paraffin sections were placed onto thermo superfrost slides (Thermo Fisher Scientific, USA). For staining, tissues were deparaffinized and blocked with 5% horse serum in 0.3% Triton X-100 for 1 h at room temperature. Then, the slides were incubated overnight (4 °C) with primary antibodies: anti-CD326 (1:100, Acris antibodies, San Diego, CA, USA; #AM33039PU-N), anti-cytokeratin (1:400, Biolegend, San Diego, CA, USA; #628602), anti-Lyve1 (1:20, R&D Systems, Minneapolis, MN, USA; #AF 2125), anti-endomucin (1:50, eBioscience, San Diego, CA, USA; #14–5851-85), anti-CD11b (1:200, Abcam, Cambridge, UK; #ab 75476), anti-F4/80 (1:100, eBioscience, San Diego, CA, USA; #14-4801). The next day, the slides were washed with 0.3% Triton X-100 in phosphate-buffered saline (PBS) and incubated with corresponding Alexa fluor secondary antibodies (room temperature, 2 h). Then, the slides were washed and mounted in Hoechst 33347 mounting medium (Sigma–Aldrich, Steinheim Germany). Images were generated using a Leica confocal microscope (Leica Microsystems, Wetzlar, Germany) and were analyzed using ImageJ software version 1.53c (NIH, MD, USA).

### 4.5. Tumor Digestion and FACS Analysis

Mid-sized tumors were harvested for FACS analysis as described [25]. Briefly, single cell suspensions were created with the GentleMACS dissociator and the murine tumor dissociation kit (both from Miltenyi Biotec, Bergisch Gladbach, Germany) following standard procedures. Red blood cells were removed by lysis, and non-specific antibody binding was blocked with FcR blocking reagent (Miltenyi Biotec). Next, cells were stained with conjugated antibodies (anti-CD3-PE-CF594, #562286; anti-CD11b-BV605, # 563015; anti-CD31-PE-Cy7, #561410; anti-CD326-BV711, #563134; anti-Ly-6C-PerCP-Cy5.5, #560525; anti-Ly-6G-APC-Cy7, #560600; anti-NK1.1-BV510, #563096 and anti-CD146-AlexaFluor488, #562229 (all BD biosciences), anti-F4/80-PE-Cy7, #123114; anti-CD90.2-PE, #130-102-489; anti-CD146-AlexaFluor488, anti-CD206-FITC, #141704 (all Biolegend), and anti-CD45-Vioblue (Miltenyi Biotec, #130-118-953). Samples were acquired with a LSRII/Fortessa flow cytometer (BD Biosciences, Heidelberg, Germany) and analyzed using FlowJo software 7.6.1 (Treestar, Ashland, OR, USA) or FACSDiva (BD). All antibodies and secondary reagents were titrated to determine optimal concentrations. CompBeads (BD) were used for single color compensation to create multi-color compensation matrices. For gating, fluorescence minus one (FMO) controls were used. The instrument calibration was controlled daily using Cytometer Setup and Tracking beads (BD). 

### 4.6. RNA Isolation and Quantitative Real-Time PCR (RT-qPCR**)**

Total RNA was extracted from cells and primary tumor tissues using QIAgen RNeasy kits (Invitrogen, Carlsbad, CA, USA) following the manufacturer’s protocol, and RNA (1 μg) was used for reverse transcription (RTase SuperScript III, Invitrogen). Gene-specific primers (BioSpring GmbH, Frankfurt, Germany) were designed using primer BLAST and the cDNA was amplified using SensiFAST SYBR No-ROX kit (GENTAUR GmbH, Aachen, Germany) and a real-time thermal cycler (Agilent Technologies Mx3000, Waldbron, Germany). All RNAs were normalized to 18S rRNA (Table 1).

### 4.7. Immunoblotting

Samples were lysed in Triton X-100 buffer and detergent-soluble proteins were solubilized in SDS-PAGE sample buffer, separated by SDS-PAGE, and subjected to Western blotting as described [47]. Proteins were visualized by enhanced chemiluminescence using a commercially available kit (Amersham, Freiburg, Germany). The antibodies against CPT1A (1:1000, abcam; #ab128568), sEH (1:5000, Cayman Chemical, Ann Arbor, MI, USA; #10010146), and β-actin (1:1000, Sigma, #A1978) were used.

### 4.8. UPLC–MS/MS-Based Fatty Acid Metabolite Profiling 

For the preparation of calibration curves, stock solutions were prepared in ethanol that contained primary fatty acids and oxylipin standards. Working standard solutions were prepared by serial dilution of the stock solutions to create the necessary concentrations. A solution containing 22 deuterium-labeled internal oxylipin standards and ^13^C-labeled arachidonic acid was prepared at 0.1 ng/μL in ethanol. All solutions were stored at −80 °C until use.

All samples were spiked with 10 µL internal standard stock solution. Lipids were extracted using two-phase liquid–liquid extraction in which ethyl acetate (500 µL) was added, samples were rigorously vortexed, centrifuged (10,000× *g*, 5 min, 4 °C), and the upper organic phase was collected. The extraction was repeated. These lipid extracts were evaporated to dryness under a continuous nitrogen stream (Vacuum manifold, Macherey-Nagel, Düren, Germany) and subsequently resuspended in 50/50 methanol/water (containing 100 ng/mL flufenamic acid as internal control). Finally, samples were analyzed by UPLC–MS/MS for primary fatty acids and oxylipins.

The UPLC–MS/MS analyses were performed as described [48], with some minor modifications. Reversed-phase separation was performed on an Acquity UPLC BEH shield RP18 column (2.1 × 100 mm; 1.7 μm; Waters, Milford, MA, USA) on an Agilent 1290 Infinity LC system (Agilent, Waldbronn, Germany). The mobile phase consisted of (A) ACN/water/acetic acid (60/40/0.02, *v*/*v*/*v*) and (B) ACN/isopropanol (50/50, *v*/*v*). Elution of analytes was carried out for 5.8 min at a flow rate of 0.5 mL/minute. Gradient conditions were as follows: 0–4.5 min, 0.1–55% B; 4.5–5.0 min, 55–99% B; 5.0–5.8 min, 99% B, followed by a re-equilibration step 0.1% B for 2 min. Then, 8 μL of each sample was injected onto the column. The column temperature was kept at 40 °C. Samples were kept at 4 °C until analysis. Mass spectrometry was performed on a QTrap 5500 mass spectrometer (Sciex, Darmstadt, Germany), equipped with a Turbo V ion source. Electrospray ionization in negative mode was employed. The ion source parameters were as follows: CUR 20 psi, IS -4500 V, TEM 525 °C, GS1 30 psi, GS2 30 psi, CAD medium (nitrogen was employed as the collision gas). Analyst 1.6.2 and MultiQuant 3.0 (Sciex, Darmstadt, Germany) were used for data acquisition and analysis, respectively.

### 4.9. Proteomics: Sample Preparation

*Tumor lysates*: Primary tumors were solubilized in 10% SDS (*w*/*v*), 150 mmol/L NaCl, 100 mmol/L HEPES pH 7.8. Samples were sonicated for 5 s, heated for 5 min at 95 °C, and centrifuged to remove insoluble material. Total protein amount of 100 µg of each sample was diluted in (4% (*w*/*v*) SDS, 100 mmol/L HEPES, pH 7.6, 150 mmol/L NaCl, 0.1 M DTT, mixed with 200 µL 8 mol/L urea, 50 mmol/L Tris/HCl, pH 8.5 and loaded to spin filters with a 30 kDa cut-off (Microcon, Merck, Darmstadt, Germany). The filter-aided sample preparation protocol (FASP) was followed as described [49]. Proteins were digested overnight with trypsin (sequencing grade, Promega, Mannheim, Germany). Acidified peptides (final concentration 0.1% *v*/*v* trifluoroacetic acid) were fractionated on multi-stop-and-go tips (StageTips) containing C18 and strong cation exchange (SCX) tips (Empore, St. Paul, MN, USA), as outlined [50]. The C18 translution fraction was combined with the first SXC fraction. All six fractions of each sample were eluted in wells of microtiter plates. Peptides were dried and resolved in 1% acetonitrile, 0.1 % formic acid.

### 4.10. Proteomics: Mass Spectrometry

LC–MS was performed on Thermo Scientific™ Q Exactive Plus equipped with an ultra-high performance liquid chromatography unit (Thermo Scientific Dionex Ultimate 3000) and a Nanospray Flex Ion-Source (Thermo Scientific). Peptides were loaded on a C18 reversed-phase precolumn (Thermo Scientific) followed by separation on a 2.4 µm Reprosil C18 resin (Dr. Maisch GmbH, Ammerbuch-Entringen, Germany) in-house packed picotip emitter tip (diameter 100 µm, 30 cm long from New Objectives) using a gradient from mobile phase A (4% acetonitrile, 0.1% formic acid) to 30% mobile phase B (80% acetonitrile, 0.1% formic acid) for 100 min following a second gradient to 60% B for another 45 min with a flow rate of 200 nL/minute. 

MS data were recorded by a data-dependent acquisition method selecting the ten most abundant precursor ions in positive mode for HCD fragmentation. Lock mass option [51] was enabled to ensure high mass accuracy during many following runs. The full MS scan range was 300 to 2000 m/z with resolution of 70,000, and an automatic gain control (AGC) value of 3 × 10^6^ total ion counts with a maximal ion injection time of 160 milliseconds. Higher charged ions (2+) were selected for MS/MS scans with a resolution of 17,500, an isolation window of 2 m/z, and an automatic gain control value set to 10^5^ ions, with a maximal ion injection time of 150 milliseconds. Selected ions for MS/MS were excluded in a time frame of 30 s. 

### 4.11. Data Processing 

X calibur Raw files were analyzed by proteomics software Max Quant (1.6.4.0) [52]. The enzyme specificity was set to trypsin, and missed cleavages were limited to 2. Acetylation of *N*-terminus (+42.01), deamidation on N and Q, and oxidation of methionine (+15.99) were selected for variable modification; carbamidomethylation (+57.02) on cysteines was set as fixed modification. Mouse proteome set from Uniprot (Download February 2018, 52,538 entries) was used to identify peptides and proteins with a false discovery rate (FDR) of 1%. Label-free quantification values were obtained from at least one identified peptide. Identifications from reverse decoy database, by site identifications and known contaminants, were excluded. Data were further statistically analyzed by Perseus 1.6.1.3 [53] and Microsoft Excel 2013. Quantified proteins in tumor lysates were quality filtered according to a minimum of six valid values in one experimental group (*N* = 7). Missing values were replaced by random background values from normal distribution. For statistical comparison, Student’s *t* tests and permutation-based FDR were used. The mass spectrometry proteomics data have been deposited in the ProteomeXchange Consortium via the PRIDE [54] partner repository with the dataset identifier PXD026214. 

### 4.12. Statistical Analysis

Data are expressed as mean ± SEM. Statistical evaluation was performed using Student’s *t* test for unpaired data or ANOVA for repeated measures followed by Newman–Keuls comparison test where appropriate. Values of *p* < 0.05 were considered statistically significant.

## Figures and Tables

**Figure 1 ijms-22-07120-f001:**
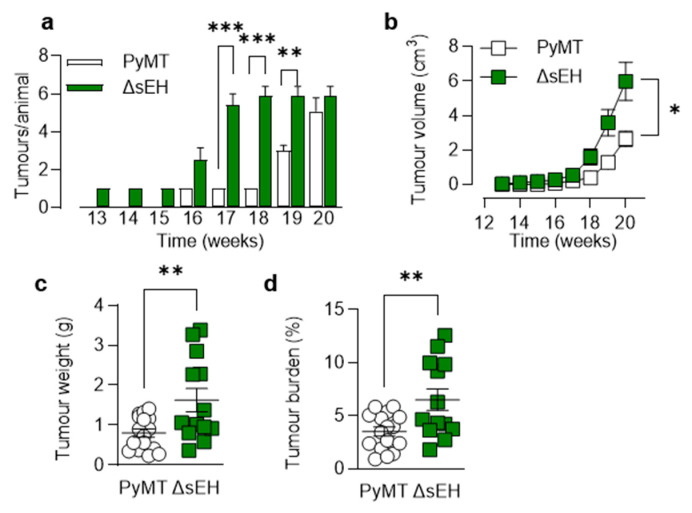
Consequences of sEH deletion on primary tumor growth. (**a**) Time-dependent increase in tumor number per animal from weeks 13 to 20; *n* = 8–13 animals per group. (**b**) Tumor volume per animal/week from weeks 13 to 20; *n* = 8–13 animals per group. (**c**) Total tumor weight at week 20; *n* = 13–19 animals per group. (**d**) Tumor burden (total tumor weight normalized to body weight) at week 20; *n* = 13–19 animals per group. * *p* < 0.05, ** *p* < 0.01, *** *p* < 0.001 ((**a**,**b**) ANOVA for repeated measures and Newman–Keuls test, (**c**,**d**) Student’s *t* test).

**Figure 2 ijms-22-07120-f002:**
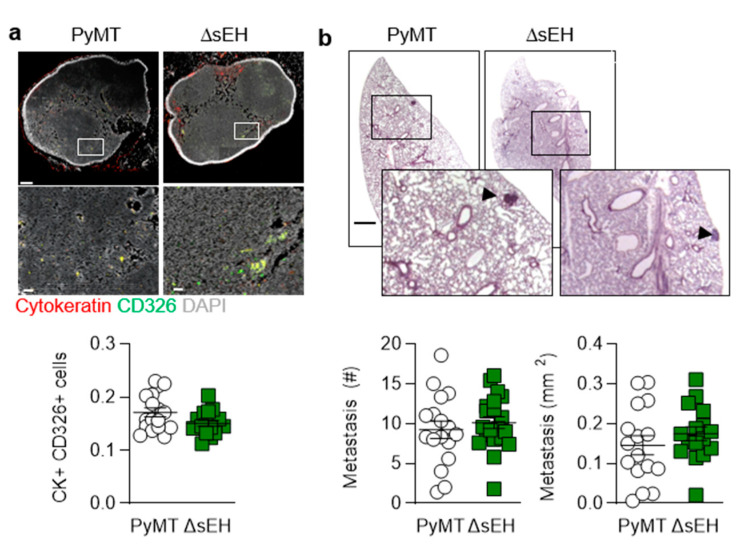
Lymph node and pulmonary metastases. (**a**) Metastatic tumor cells identified using cytokeratin (red) and CD326 (green) in axillary lymph nodes from PyMT and PyMT^ΔsEH^ (ΔsEH) mice, bar = 200 µm (upper panels) 20 µm (lower panels); *n* = 16–17 animals per group. (**b**) Breast cancer metastases (HE staining), metastasis number and size in lungs from PyMT and PyMT^ΔsEH^ (ΔsEH) mice; bar = 1 mm. The inserts show magnifications of the areas marked by boxes; *n* = 16 animals per group with 5 sequential slides evaluated per sample.

**Figure 3 ijms-22-07120-f003:**
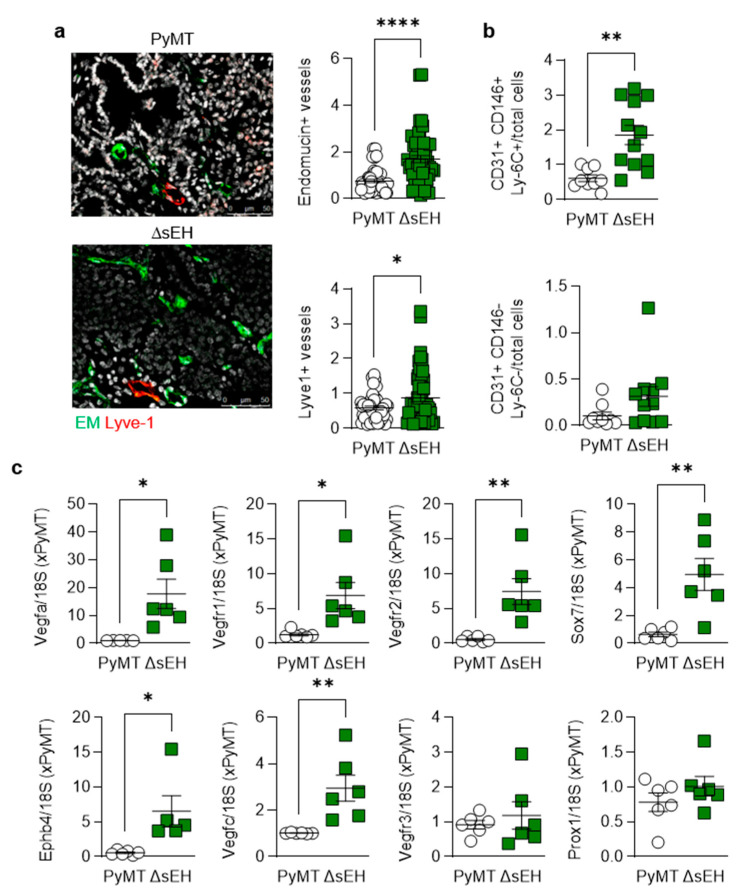
Angiogenesis and lymphangiogenesis in primary tumors. Primary tumors were removed from 20-week-old PyMT and PyMT^ΔsEH^ (ΔsEH) mice. (**a**) Vascular endothelial cells; endomucin+ (EM, green) and lymph endothelial cells (Lyve-1+, red); bar = 50 µm. *n* = 8–12 animals per group with 5 sequential slides evaluated per tumor. (**b**) FACS analysis of primary tumor digests for endothelial cells (CD31+, CD146+, Ly-6C+) and lymph endothelial cells (CD31+, CD146-, Ly-6C-); *n* = 9–13 animals per group. (**c**) Expression of angiogenesis markers (VEGFA, VEGFR1, VEGFR2, SOX7, EphB4) and lymph angiogenesis markers (VEGFC, VEGFR3, and Prox1) in primary tumors; *n* = 6 animals per group. * *p* < 0.05, ** *p* < 0.01, **** *p* < 0.0001 (Student’s *t* test).

**Figure 4 ijms-22-07120-f004:**
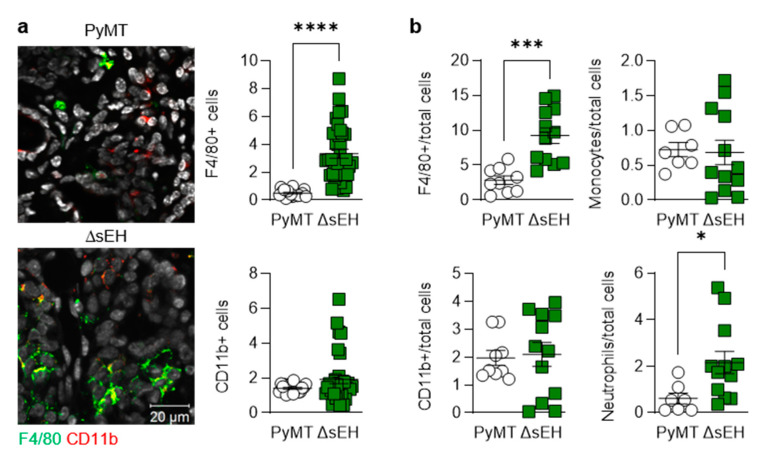
Lymphocyte infiltration into primary tumors. Primary tumors were removed from 20-week-old PyMT and PyMT^ΔsEH^ (ΔsEH) mice. (**a**) Macrophages; F4/80+ (green) and leukocytes; CD11b+ (red). *N* = 8–12 animals per group with 5 sequential slides evaluated per tumor. (**b**) FACS-based quantification of primary tumor digests for macrophages (F4/80+), leukocytes (CD11b+), monocytes (Ly-6C+), and neutrophils (Ly-6G+). *N* = 9–12 animals per group. * *p* < 0.05, *** *p* < 0.001, **** *p* < 0.0001 (Student’s *t* test).

**Figure 5 ijms-22-07120-f005:**
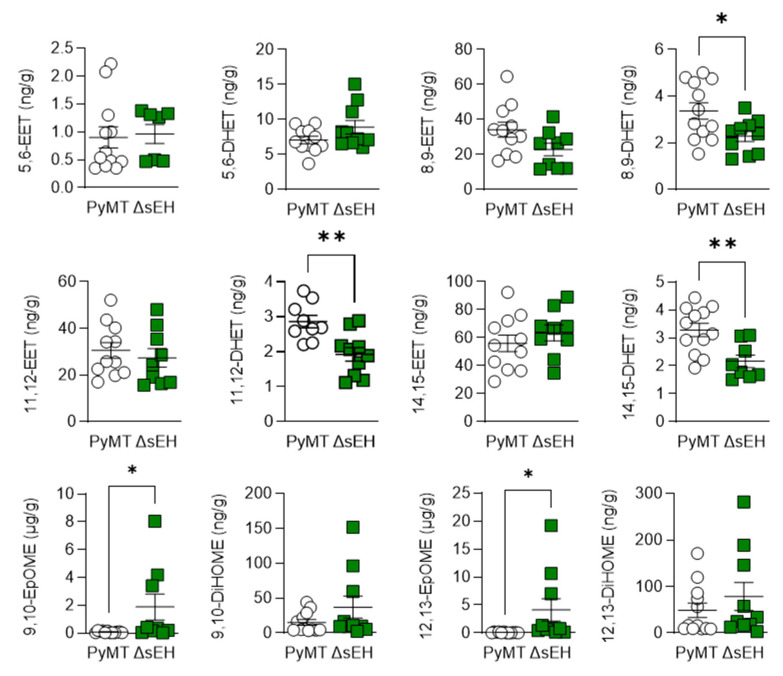
ω-6 PUFA epoxide and diol profiles. Primary tumors were isolated from 20-week-old PyMT and PyMT^ΔsEH^ (ΔsEH) mice and the PUFA epoxide and diol profiles were generated by LC–MS/MS. Note the different scale in the X axes of the EpOMEs. EET, epoxyeicosatrienoic acid; DHET, dihydroxyeicosatrienoic acid; EpOME, epoxyoctadecamonoenic acid; DiHOME, dihydroxyoctadecenoic acid. The graphs summarize data from breast tumors from 6–12 different animals in each group; * *p* < 0.05, ** *p* < 0.01 (Student’s *t* test).

**Figure 6 ijms-22-07120-f006:**
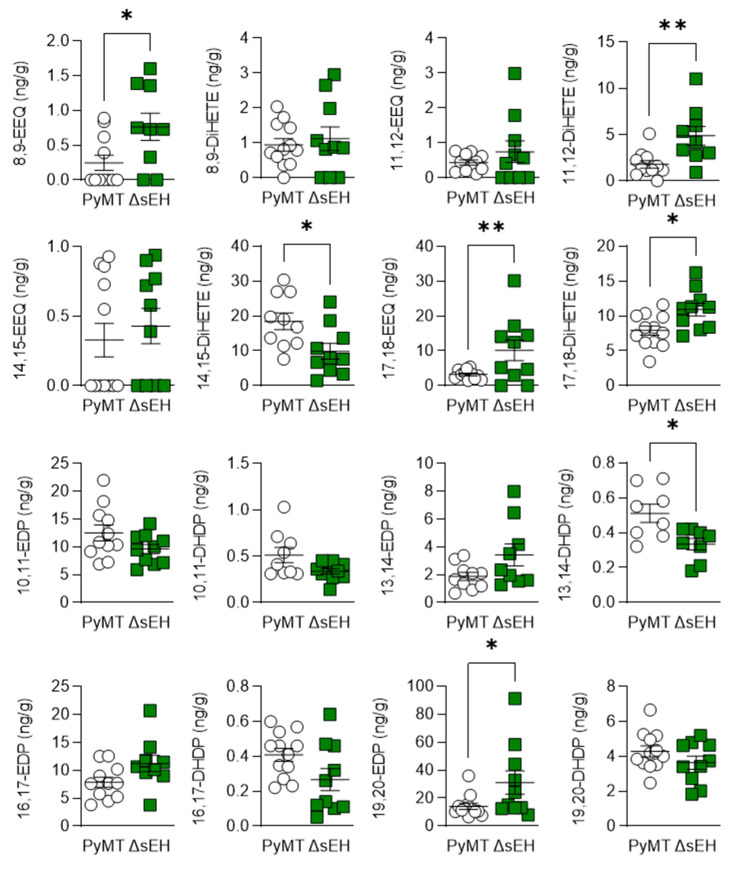
ω-3 PUFA epoxide and diol profiles. Primary tumors were isolated from 20-week-old PyMT and PyMT^ΔsEH^ (ΔsEH) mice and the PUFA epoxide and diol profiles were generated by LC–MS/MS. EEQ, epoxyeicosatetraenoic acid; DiHETE, dihydroxyeicosatetraenoic acid; EDP, epoxydocosapentaenoic acid; DHDP, dihydroxydocosapentaenoic acid. The graphs summarize data from breast tumors from 10–12 different animals in each group; * *p* < 0.05, ** *p* < 0.01 (Student’s *t* test).

**Figure 7 ijms-22-07120-f007:**
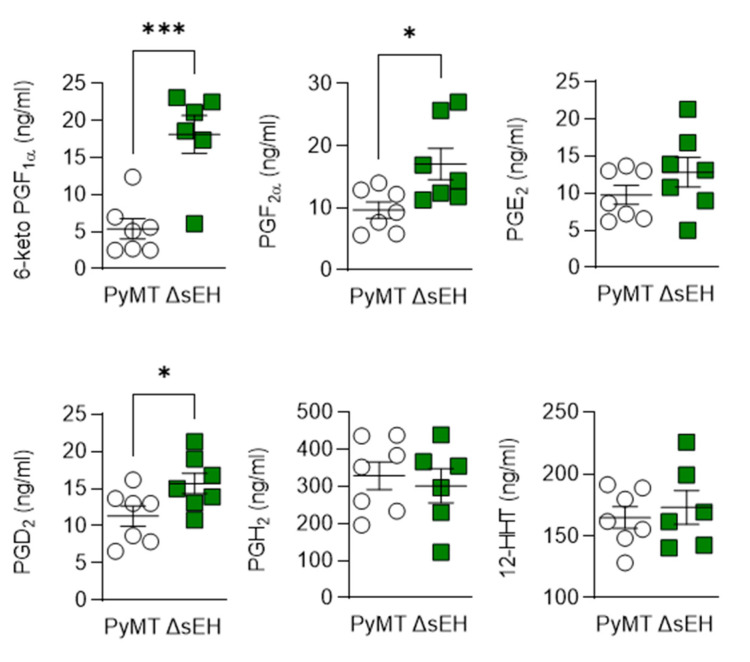
Consequences of Cyp2c44 and sEH deletion on the prostaglandin profile in breast cancer tumors. Tumors were isolated from 20-week-old PyMT and PyMT^ΔsEH^ (ΔsEH) mice and prostaglandin (PG) levels assessed by LC–MS/MS. The graphs summarize data from *n* = 6–7 different animals in each group. * *p* < 0.05, *** *p* < 0.001 (Student’s *t* test).

**Figure 8 ijms-22-07120-f008:**
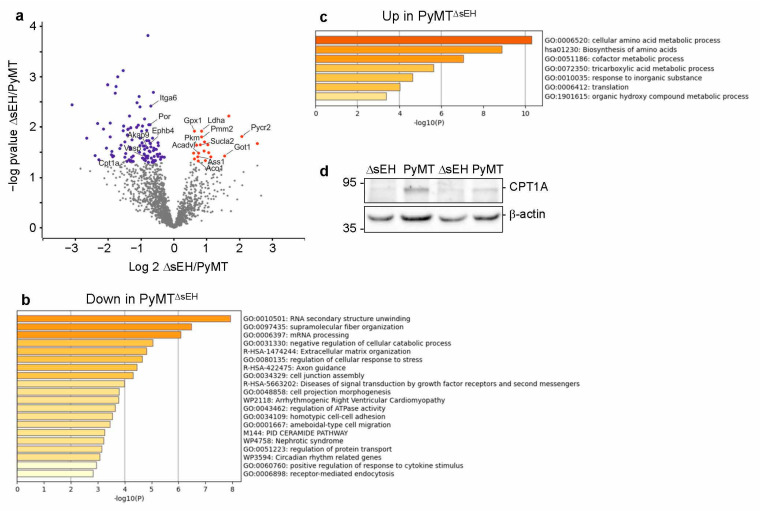
Altered protein expression in primary tumors from PyMT and PyMT^ΔsEH^ mice. (**a**) Volcano plot showing the proteins most altered in tumors from PyMT^ΔsEH^ (ΔsEH) versus PyMT mice. (**b**,**c**) Pathway analysis with enrichment scores for GO term annotations showing proteins downregulated (**b**) or upregulated (**c**) in sEH-deficient tumors. (**d**) CPT1A expression in primary tumors from PyMT and PyMT^ΔsEH^ mice.

**Table 1 ijms-22-07120-t001:** Primers used for RT-qPCR.

Gene Name (Mouse)	Accession Number	Forward Primer	Reverse Primer
18S	NR_003278.3	CTTTGGTCGCTCGCTCCTC	CTGACCGGGTTGGTTTTGAT
Prox1	NM_008937.3	CCAGTAAGACATCACCGCGT	TGGGCACAGCTCAAGAATCC
Sox7	NM_011446.1	AAACCCCACTCTGGCTTGAC	GGTCCTTGGGCAGTCATTCA
Vegfr1	NM_010228.4	GAGGAGGATGAGGGTGTCTATAGGT	GTGATCAGCTCCAGGTTTGACTT
Vegfr2	NM_010612.3	GCCCTGCTGTGGTCTCACTAC	CAAAGCATTGCCCATTCGAT
Vegfr3	NM_008029.3	ATGTGTGGTCCTTCGGCGTGC	TTCAGCCGCTGGCAGAACTCC
Vegfa	NM_001287056.1	GCACTGGACCCTGGCTTTACT GCTGTA	GAACTTGATCACTTCATGGGACTTCTGCTC
Vegfc	NM_009506.2	GTGCTTCTTGTCTCTGGCGT	TTCAAAAGCCTTGACCTCGC
EphB4	NM_001159571.1	GGCGCATTGGGTTTCTTTCT	CTGGGGATAGCCCATGACAG

## Data Availability

The original data generated during the current study are available from the corresponding author on reasonable request. The mass spectrometry proteomics data (**Dataset 1**) have been deposited in the ProteomeXchange Consortium via the PRIDE [54] partner repository with the dataset identifier PXD026214.

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
