# Peer review of "The Consequences of Soluble Epoxide Hydrolase Deletion on Tumorigenesis and Metastasis in a Mouse Model of Breast Cancer"

_ijms, 2021, doi:10.3390/ijms22137120_

Round 1

Reviewer 1 Report

This manuscript described the relationship between the sEH deletion and tumorigenesis/metastasis. Experiments are well designed and appropriately analyzed. There are few comments that should be addressed: (1) details of the establishment of the PyMT-delta-sEH line are required to justify the equivalence of the genetic backgrounds between the two lines; (2) elaboration of the results of the proteome analysis would help understanding its significance; and (3) L234-237 “sEH deletion … failed to stimulate lymphangiogenesis” is somewhat misleading.

Author Response

Reply: We would like to thank the reviewers for her/his constructive criticism of our manuscript. We hope that the additional data provided and the changes made are sufficient to allay the concerns raised.

(1) Details of the establishment of the PyMT-delta-sEH line are required to justify the equivalence of the genetic backgrounds between the two lines.

Reply: As outlined in the methods section Polyoma virus middle T-310 antigen (PyMT) mice (FVB/N-Tg(MMTV-PyVT)634Mul/J) mice were purchased from The Jackson Laboratory (Bar Harbor, ME) and then crossed with sEH-/- mice to generate PyMT mice lacking sEH (PyMTDsEH).

(2) elaboration of the results of the proteome analysis would help understanding its significance.

Reply: We appreciate the reviewers comment. We have added some additional comments as requested but have been careful to keep the balance between reporting and over speculation.

 (3) L234-237 “sEH deletion … failed to stimulate lymphangiogenesis” is somewhat misleading.

Reply: We appreciate the reviewers point, our comment was meant to refer to the differences between the PyMTDsEH mice and PyMT mice lacking Cyp2c44, which had a very pronounced lymphangiogenesis and metastasis. We have reworded the sentence to be clearer. The revised sentence now reads “Perhaps the differential effect on metastasis can be accounted for by the fact that sEH deletion, in contrast to the deletion of Cyp2c44, did not markedly stimulate lymphangiogenesis [16], which seems to be required for metastasis in PyMT mice [25].”

Reviewer 2 Report

Modification polyunsaturated fatty acids (PUFAs) can influence many biological processes such as tumor cell proliferation and angiogenesis. One of the important modifications is through the function of epoxide hydrolases. In mammals, there are soluble epoxide hydrolase (sEH), microsomal epoxide hydrolase (mEH), cholesterol epoxide hydrolase, and leukotriene A4 (LTA4) hydrolase. These enzymes mediate the addition of water to both exogenous and endogenous epoxides, leading to the corresponding vicinal diols.  

This is a very interesting paper in which the authors investigated the consequences of PUFA epoxides in regards to metastases in a model of spontaneous breast cancer using sEH-/- mice. The experimental design provided more physiological condition from previous studies in mice which were performed  by injecting cancer cells into animals overexpressing the human CYP2C8 or CYP2J2 enzymes in endothelial cells and a subset (Tie2-expressing) of myeloid cells, or following the  systemic administration of high concentrations of 14,15-EET. However the authors should provide some more detailed analysis in the study.

  • In mammals, there are soluble epoxide hydrolase (sEH), microsomal epoxide hydrolase (mEH), cholesterol epoxide hydrolase, and leukotriene A4 (LTA4) hydrolase. These enzymes mediate the addition of water to both exogenous and endogenous epoxides, leading to the corresponding vicinal diols.   Often times, there are compensatory mechanisms for homeostatic control of biological processes.

The authors should assay for changes in expression and activity of microsomal epoxide hydrolase (mEH), cholesterol epoxide hydrolase, and leukotriene A4 (LTA4) hydrolase in the tissues of the sEH-/- mice. These data would provide a more detailed picture for the total changes of the epoxide hydrolase activity in the tissue studied.

  • In the PyMTDsEH mice, breast cancer development was studied in mice expressing the polyoma middle T oncogene (PyMT) under the control of the mouse mammary tumor virus promoter, to induce spontaneous mammary tumors.  The level and activity of sEH in various tissues of PyMT should be provided to demonstrate the expression of sEH with the induction of spontaneous mammary tumors.

  • For the genes in Table 1, the accession # should be provided.

  • Please provide the catalogue # for the antibodies used in the study.

Author Response

We would like to thank the reviewers for her/his constructive criticism of our manuscript. We hope that the additional data provided and the changes made are sufficient to allay the concerns raised.

  1. In mammals, there are soluble epoxide hydrolase (sEH), microsomal epoxide hydrolase (mEH), cholesterol epoxide hydrolase, and leukotriene A4 (LTA4) hydrolase. These enzymes mediate the addition of water to both exogenous and endogenous epoxides, leading to the corresponding vicinal diols. Often times, there are compensatory mechanisms for homeostatic control of biological processes.

Reply: Compensatory increases in the mEH or the LTA4 hydrolase were not detected in sEH-/- mice (in fact there was a slight tendency for mEH pexpression to also decrease). This was evident from the proteomics dataset but the data have been reproduced in Figure A for the reviewers benefit.

  1. The authors should assay for changes in expression and activity of microsomal epoxide hydrolase (mEH), cholesterol epoxide hydrolase, and leukotriene A4 (LTA4) hydrolase in the tissues of the sEH-/- mice. These data would provide a more detailed picture for the total changes of the epoxide hydrolase activity in the tissue studied.

Reply: Given as we only have frozen or fixed tissue available we have not included the requested activity assays – which are best performed in fresh tissue. This comment is related to the previous one – and as we found the expression of the mEH and LTA4 hydrolase was comparable in the 2 genotypes (see Figure A) we feel confident in stating that there were no differences. Cholesterol epoxide hydrolase was not detected in the proteome of tumors from PyMT and PyMTDsEH mice.

In the PyMTDsEH mice, breast cancer development was studied in mice expressing the polyoma middle T oncogene (PyMT) under the control of the mouse mammary tumor virus promoter, to induce spontaneous mammary tumors. The level and activity of sEH in various tissues of PyMT should be provided to demonstrate the expression of sEH with the induction of spontaneous mammary tumors.

Reply: If we understand the reviewers comment correctly, she/he is asking us to look for the expression of the sEH in different tissues during the period of spontaneous tumor development. Unfortunately, the revision time allowed (10 days) means that we are unable to go back and assess sEH activity is tissues from mice of different ages. However, we have provided western blots showing the expression of the sEH in primary tumors from PyMT mice.

For the genes in Table 1, the accession # should be provided.

Reply: Accession numbers have been added to Table 1 as requested.

Please provide the catalogue # for the antibodies used in the study.

Reply: Catalogue numbers have been provided for the antibodies used, as requested.

Round 2

Reviewer 2 Report

No more concerns from the reviewer.